# Seasonal Variation in the Body Composition, Carcass Composition, and Offal Quality in the Wild Fallow Deer (*Dama dama* L.)

**DOI:** 10.3390/ani13061082

**Published:** 2023-03-17

**Authors:** Marek Stanisz, Maciej Skorupski, Marta Bykowska-Maciejewska, Joanna Składanowska-Baryza, Agnieszka Ludwiczak

**Affiliations:** 1Department of Animal Breeding and Product Quality Assessment, Faculty of Veterinary Medicine and Animal Science, Poznan University of Life Sciences, Sloneczna 1, 62-002 Zlotniki, Poland; 2Department of Game Management and Forest Protection, Faculty of Forestry, Poznan University of Life Sciences, Wojska Polskiego 28, 60-637 Poznań, Poland; maciej.skorupski@up.poznan.pl

**Keywords:** wild ungulates, hunt-harvested deer, season, edible internal organs

## Abstract

**Simple Summary:**

Game meat popularity is connected with the fact that harvesting game is part of the sustainable management of game species. As hunted animals grow in their natural environments—with access to natural forage, exercise, and unlimited possibilities to perform natural behaviors—consumers are enabled to think about game meat as a healthy product, obtained from animals that grew in welfare-friendly conditions. On the other hand, the popularity of using offal (including those from game species) in dishes or processed products is lower compared to muscle tissue and differs among cultures and regions. Nevertheless, we should remember that offal is a nutritional and valuable animal product, and its use allows us to limit the amounts of utilized by-products, and therefore should also be considered from the perspective of sustainability in the meat production chain and its environmental effects.

**Abstract:**

The goal of this study was to examine the seasonal variation in the body composition, carcass composition, and quality of edible internal organs from the fallow deer hunt-harvested in the summer (*n* = 9) and the winter (*n* = 10) seasons. The weight and proportion of the mesenteric and omental fat were greater for the animals harvested in the winter (1.12 kg and 2.75%) compared to those from the summer season (0.43 kg and 1.02%). The winter-harvested animals had more perinephric fat (0.75 kg and 1.84%) than those hunted in summer (0.26 kg and 1.84%). The gastrointestinal tract of the fallow deer hunted in summer was more filled with feed and therefore heavier (7.92 kg) compared to those from the winter season (5.16 kg). The proportion of fat was significantly greater in the carcasses obtained in winter compared to the summer season (6.55% vs. 3.79%). No seasonal variety was found in the physicochemical characteristics of the edible offal, but the content of extractable fat was significantly affected by the season. In conclusion, the effect of the season on the slaughter value of the hunt-harvested fallow deer was limited to a variation in the proportion of some of the internal organs and affected the fat deposition in the body of the examined animals. The season significantly affected the fat content in the carcass and the extractable fat content in the examined offal.

## Disclaimer: 

This paper includes a section on the nutritional quality of wild deer offal. The internal organs of wild animals, particularly the liver and digestive tract, are at high risk of containing parasites (or pathogens), therefore constituting a health risk to humans if infected animals are not efficiently detected. This is the reason why, in several countries, the offal consumption of wild animals is forbidden. Under Regulation 1703 (2015) of the Ministry for Agriculture and Rural Development of Poland, this is allowed, and Regulation (EC) No 853/2004 of the European Parliament and Council does not prohibit it. However, by this disclaimer note, we call the attention of the readers that offal consumption from wild deer may constitute a health risk.

## 1. Introduction

Body composition and weight are good indicators of energy status in wild and farmed cervids, and both of these traits are affected by both the season and the age of the animals [1,2,3,4]. The fat depots in the body of farmed and wild animals, including the mesenteric, omental, and perinephric fat, are also commonly used to assess the body condition [5,6]. Research focusing on the effect of the seasons on the body composition of wild ungulates is of significant importance as the proportion of the most valuable cuts and the not-edible parts (skin, head, and feet) determines the profitability of this special branch in the meat industry [7,8,9,10,11]. According to the available research, the seasonal changes in the body condition of wild animals are related to the photoperiod and feed availability, with the second factor strictly dependent upon the type of environment and the population density [12,13,14,15]. Venison is valued as a culinary meat in different parts of the world [16,17,18,19], but its popularity in the production of processed products is also increasing [20]. Meat processing gives the possibility of utilizing edible animal offal and leads to the production of shelf-stable meat products. The most valuable animal offal used in meat processing, as well as for culinary purposes, are the liver, kidneys, heart, and tongue [21,22,23,24]. The hunting seasons for game species vary among European countries. In Poland, the hunting season is defined by the Directive of the Ministry of Environment [25], and for fallow deer females, it ranges between 1 September and 15 January.

The goal of our study was to examine the seasonal differences in the body composition, carcass composition, and offal quality in wild fallow deer. The summer and winter seasons were selected in this study to contrast the poorest and the best season with regard to food availability in wildlife. This study should be interesting to game processors, as depending on the slaughter value and offal quality, the processing chain may vary.

## 2. Materials and Methods

In this study, the hunters obtained an additional agreement from the Ministry of the Environment (Agreement no. DL-III.6713.15.2016) to harvest fallow deer in the summer season (from 1 August till 30 September) for scientific purposes. The decision was supported by the great number of heads of this species and the lack of participation in the reproduction of two year old animals. According to Polish legislation, hunters are responsible for the assessment of the health of hunted deer by means of the inspection of the internal organs during the act of dressing out the animal in the field [26]. Game carcasses and their edible offal can be obtained through direct sail and can be consumed [26].

### 2.1. Animals and Acquisition of Edible Offal

The research material was 19 two year old fallow deer does (bled body weight of 40–41 kg), hunt-harvested in the same hunting area of western Poland in the summer season (July; *n* = 9) and in the winter (December; *n* = 10). The fallow deer in this study were harvested in accordance with the requirements of the Polish Hunting Law implemented by the Government of Poland. The age was determined by a trained hunter after the shot through the replacement and the level of wear of the premolars and molars. Only animals in their second year of life were intended for use in this study. The animals were shot during a snipe hunt (in the head or in the neck), bled out, and weighted (referred to as the bled body weight). Next, the deer were hung by the Achilles tendon. The hot carcass weight was recorded after the removal of the skin, head, internal organs, and feet. The following non-edible carcass parts and internal organs were weighted: head, skin, front and hind feet, gastrointestinal tract, mesenteric and omental fat, perinephric fat, edible offal. These were weighted separately, and all of the internal organs were weighted together (trachea, lung, heart, liver, diaphragm, kidneys, spleen). Additionally, in order to examine the effect of the season on the edible offal quality, the liver, kidneys, heart, and tongue were weighed separately and defined with the term: point of measurement (PM). All of these procedures were conducted in a game collection point. After evisceration, the carcasses were transported to a slaughterhouse in a meat truck, under chilled conditions (+2 °C). After 24 h from the hunting, the deer carcasses were halved and the half-carcasses were jointed according to the methodology proposed by Stanisz et al. [24] into cuts: neck, shoulder, loin, ribs with flank, and leg. The carcass parts were dissected into meat, fat, and bones, and the proportion of these tissues in the carcass and carcass parts was calculated. The edible offal was kept at +2 °C during storage.

### 2.2. Physicochemical Traits of Edible Offal

The pH of the edible internal organs was measured with a temperature-compensated portable pH-meter, pH 1140 (Mettler Toledo, Urdorf, Switzerland) equipped with a combination glass calomel electrode (Lo 406-M6-DXK-S7/25). Before the measurement, the pH equipment was calibrated with 4.0 and 7.0 buffers.

The water holding capacity (WHC, %) was measured according to the method of Grau and Hamm [27] in modification of Pohja and Niinivaara [28]. Samples (0.300 g) of ground material were placed on filter paper between two glass tiles. A force of 2 kg was applied to each sample for 5 min. Then, the samples were removed from the filter paper and reweighed immediately to calculate the change in their weight, referred to as free water. Plasticity (cm^2^) measurement was conducted according to the method of Pohja and Niinivaara [28], simultaneously with the free water measurement. The plasticity was expressed as the area of the pressed sample, marked on the filter paper, scanned, and measured using the ImageJ ver. 1.52 g software.

The examination of the chemical composition of the offal was conducted 24 h postmortem, according to the AOAC methodology [29]. The analysis included the determination of the dry matter content, total protein content with the Kjeldahl procedure (K-424 Buchi digestion unit; Büchi Labortechnik AG, Flawil, Switzerland), extractable fat content by Soxhlet extraction (Soxtec^TM^ 8000, FOSS, Warsaw, Poland), and ash content (FCF 12 SM, CZYLOK, Jastrzebie-Zdroj, Poland). On the basis of the obtained data, the content of moisture and the water-to-protein ratio were calculated.

### 2.3. Statistical Analysis

The effect of the hunting season on the body weight, carcass weight, carcass parts and offal:Y_ij_ = μ + α_i_ + e_ij_.
where:

μ—the overall mean of the analyzed trait,

α_i_—the effect of the ith season (i = 1, 2),

e_ij_—random error.

The effect of the hunting season, organs, and time *postmortem* on the pH was calculated with the model:Y_ijkl_ = μ + α_i_ + π_j_ + γ_j(k)_ + β_j(k)(l)_ + (αγβ)_ikl_ + e_ijkl_.
where:

μ—the overall mean of the analyzed trait,

α_i_—the fixed effect of the ith season (i= 1, 2),

π_j_—the random effect of jth animal (j = 1, 2, 3,…19),

γ_j(k)_—the random effects of kth organs nested in jth animal,

β_j(k)(l)_—the effect of lth time *postmortem* (l = 1, 2) as the repeated measures nested in the kth organ (k = 1, 2, 3, 4) nested in jth animal,

(αγβ)_ikl_—interaction (hunting season × organ × time *postmortem*)

e_ijkl_—the random error.

The effect of the hunting season and organs on the dry matter, crude protein, extractable fat, ash, and the W/P, moisture, WHC, and plasticity of the fallow deer was calculated according to the model:Y_ijk_ = μ + α_i_ + π_j_ + γ_j(k)_ + (αγ)_ik_ + e_ijk_.
where:

μ—the overall mean of the analyzed trait,

α_i_—the fixed effect of the ith season (i = 1, 2),

π_j_—the random effect of jth animal (j = 1, 2, 3,…19),

γ_j(k)_—the random effects of kth organ nested in jth animal,

(αγ)_ik_—interaction (hunting season × organ)

e_ijk_—the random error.

All of the statistical calculations were conducted using SAS ver.9.4 [30]. Tukey–Kramer adjustment was implemented for multiple comparisons of the Least Squares Mean (LSM) differences.

## 3. Results

### 3.1. Body Composition

The carcass characteristic of the fallow deer hunt-harvested in two seasons are given in Table 1. A higher proportion of hot and cold carcass weight was noted in the winter compared to the summer season (*p* = 0.038 and *p* = 0.029). The weight and proportion of mesenteric and omental fat (*p* = 0.001) and perinephric fat (*p* = 0.001) were also higher in the deer hunted in winter compared to those in the summer season. The weight and proportion of the full gastrointestinal tract (*p* = 0.019 and *p* = 0.016) were greater for the animals shot in the summer compared to those obtained in the winter season. The weight and proportion of the liver, kidneys, heart, tongue (Table 2), and carcass cuts did not show seasonal variation (Table 3). The composition of the fallow deer carcasses and carcass parts is presented in Table 4. The weight and proportion of fat were almost two times greater in the carcasses obtained in winter compared to those from the summer season (*p* = 0.001 and *p* = 0.001). Among the analyzed carcass cuts, the most significant seasonal variation in the fat content was observed for the shoulder, ribs with flank, and neck.

### 3.2. Physicochemical Characteristics of Edible Internal Organs

The physicochemical characteristics of the offal are presented in Table 5. The variety found in the values of these characteristics was connected with the point of measurement, but not with the season. The highest pH values were noted at 24 h and 48 h postmortem in the kidneys and heart (*p* < 0.0001). The tongue was characterized by the lowest percentage of moisture, WHC, and plasticity among the examined points of measurement (*p* < 0.0001). The highest moisture content was found in the kidneys (*p* < 0.0001). The heart was characterized by the highest percentage of WHC (*p* < 0.0001), while the liver samples revealed the greatest plasticity (*p* < 0.0001). The chemical composition of the offal from the wild fallow deer analyzed in this study differed between seasons in the extractable fat content (*p* = 0.002) (Table 6).

## 4. Discussion

### 4.1. Seasonal Variation in Body Weight and Fat Deposition

The season has a strong effect on the body weight, and the fat tissue reserves are included in a group of parameters that indicate the condition of wild animals [12,16]. In our study, we have noted the influence of the season on the hot and cold carcass weight; in both cases, it referred to the weight revealed as a percentage of the bled body weight. There was no seasonal variation in the carcass weight (both hot and cold) revealed in kilograms, similar to the results presented by Serrano et al. [18] for Iberian wild red deer. Serrano et al. [18] pointed out the effect of the season on the percentage of some cut yields in the carcass weight. In contrast to our results and those presented by Serrano [18], in the research of Ceacero et al. [16], the authors noted seasonal variation (culling in late autumn vs in late summer) in the growth performance and carcass traits in farmed fallow deer. Moreover, the literature points to a seasonally varied deposition of fat in the bodies of wild and farmed cervids. Among the fat depots, kidney fat and visceral fat are of the utmost importance [31,32]. In free-living animals, late summer and autumn are the time when the fat reserves are built up, while winter is a period of fat reserve consumption. The fluctuations of body conditions in wild cervids are affected by additional factors, such as milk production in the case of lactating does and rut in the case of bucks [16,17,33]. Ceacero et al. [16] compared fallow deer culled in late autumn and late winter and stated that the percentage of internal fat was on the same level in both groups, but the kidney fat index increased from late autumn until late winter. In the present study, a higher weight and proportion of mesenteric and omental fat and of perinephric fat were noted in the deer carcasses obtained in early winter compared to those from the late summer season. Yokoyama et al. [31] also noted seasonal changes in the amount of perinephric fat in the free-living sika deer. They found in their study that the perinephric fat weight was the greatest in August and the lowest in February. The seasonal patterns of the changes body weight and body composition in deer are associated with the changes in the percentage of body fat, protein, and water. In the winter season, the body fat and protein reserves depend on the pre-winter and post-winter levels of these components [13,34]. In the present study, the proportion of fat was significantly greater in the carcasses, as well as in the individual carcass parts, of the deer obtained in winter compared to those from the summer season. As pointed out by Volpelli et al. [35], there are additional sources of variation in the weight of the carcass and organs, such as the diet and age of the fallow deer. The available literature points out the seasonal variation in the quality of venison [15,36].

### 4.2. Effect of Season on Internal Organs and Venison

Although no effect of the season on the offal weight was found in our study, many researchers have observed seasonal fluctuations in the weight of some internal organs from wild deer. The variations in the weight of the gastrointestinal tract are caused by seasonal changes in the degree of filling the rumen with green forage [7,13,34]. This phenomenon is connected to the reduction in appetite in free-living animals and the decreased voluntary intake of feed [14,33]. The effect of the season on the weight and proportion of the full gastrointestinal tract was also observed in our study. Yokoyama et al. [31] noted a significant seasonal variation in the kidney weight in wild sika deer and explained it with energy expenditure, protein and fat catabolism, and fluctuations in the concentration of photoperiod-sensitive hormones. According to Arnold [33], the fluctuations in the weight of the liver in wild ruminants also reflect the magnitude of seasonal changes in the body condition. According to the available literature, venison quality shows a seasonal variation. Serrano et al. [18] examined the venison from Iberian wild red deer and noted that the *longissimus thoracis et lumborum* muscle from winter-harvested animals was characterized by greater pH48 (5.89 vs. 5.81; *p* < 0.001) and lower L* (36.5 vs. 38.3; *p* < 0.01) and b* (13.8 vs. 15.3; *p* < 0.01) compared to those from the autumn season. The venison from the deer hunted in autumn contained more extractable fat compared to those from the winter season (0.16 vs. 0.10 g/100 g LTL; *p* < 0.01). Stevenson et al. [7] present the rut as the major cause of seasonal variation in the venison chemical composition. The season influenced the shear force value and the winter venison was tougher than the autumn-harvested venison (20.5 vs. 18.5 N; *p* = 0.05). Stanisz et al. [14] reported that the season (winter vs summer) influences the attributes of the venison from fallow deer hunt-harvested in Poland, including the pH value, L*, a*, b*, purge loss, WHC, cooking loss, shear force, protein, and extractable fat content. The seasonal variation in the venison chemical composition was also stated by García Ruiz et al. [3]. Wiklund et al. [36] examined the quality of the venison of wild red deer and found that the WHC in venison obtained in December was greater compared to the meat of animals harvested in March, July, and September (*p* < 0.001). These findings in the quality of meat suggest that a seasonal variation in the quality of the offal can also be expected. However, the results of our study indicate that the offal physicochemical traits were not affected by the season, but by the organ itself. Some data on the physicochemical quality of the edible internal organs of animals can be found in the literature. Compared to the results in our study, slightly higher pH values in the offal from farm-raised fallow deer were given by Stanisz et al. [24]. The pH after 24 h from the slaughter measured 5.96 in the liver, 6.52 in the kidneys, 6.44 in the heart, and 5.79 in the tongue. In the study of Ludwiczak et al. [21] the pH of the offal of wild boars of different sexes and ages was measured, and amounted to 6.12–6.31 in the liver, 6.32–6.54 in kidneys, 5.81–5.98 in the heart, and 5.52–5.66 in the tongue. In general, irrespective of the species, the pH in the kidneys is higher compared to the other offal [21,22,37]. As expected on the basis of the previously presented seasonal variation in body weight and body composition, the extractable fat content in the offal from the winter harvest was greater compared to the offal of the deer obtained in the summer season.

## 5. Conclusions

To conclude, the results of this study indicate that the body composition and carcass composition of wild fallow deer show a significant seasonal variation in terms of the fat content. The animals harvested in the winter season are characterized by an increased visceral fat content and greater content of fat in the carcass compared to the deer hunted in summer. The effect of the season was also noted in the fat content of the offal from the fallow deer, but no seasonal variation in the physicochemical traits of these organs was found.

## Figures and Tables

**Table 1 animals-13-01082-t001:** Seasonal variation in the slaughter value of the fallow deer (LSM ± SE).

	Season		
Item	Summer	Winter	*p*-Value
BBW ^1^	41.8 ± 2.4	40.9 ± 3.8	0.589
Hot carcass weight, kg	23.12 ± 1.23	23.89 ± 1.22	0.628
Hot carcass weight, % BBW	55.31 ± 1.09	58.41± 1.01	0.038
Cold carcass weight, kg	22.62 ± 1.21	23.47 ± 1.19	0.537
Cold carcass weight, % BBW	54.12 ± 1.02	57.38 ± 0.95	0.029
Front feet, kg	0.62 ± 0.06	0.64 ± 0.05	0.983
Front feet, % BBW	1.49 ± 0.08	1.55 ± 0.09	0.986
Hind feet, kg	0.73 ± 0.07	0.75 ± 0.06	0.986
Hind feet, % BBW	1.75 ± 0.08	1.83 ± 0.09	0.968
Head, kg	2.16 ± 0.21	2.29 ± 0.27	0.843
Head, % BBW	5.17 ± 0.39	5.59 ± 0.38	0.965
Skin, kg	3.62 ± 0.36	3.74 ± 0.39	0.628
Skin, % BBW	8.66 ± 0.99	9.14 ± 0.98	0.243
Other organs ^2^, kg	2.93 ± 0.19	2.86 ± 0.22	0.342
Other organs ^2^, % BBW	7.02 ± 0.64	7.01 ± 0.63	0.999
Gastro-intestinal tract (full), kg	7.92 ± 0.52	5.16 ± 0.38	0.019
Gastro-intestinal tract (full), % BBW	18.95 ± 1.31	12.62 ± 1.29	0.016
Mesenteric and omental fat, kg	0.43 ± 0.12	1.12 ± 0.15	0.001
Mesenteric and omental fat, % BBW	1.02 ± 0.08	2.75 ± 0.09	0.001
Perinephric fat, kg	0.26 ± 0.07	0.75 ± 0.09	0.001
Perinephric fat, % BBW	0.63 ± 0.05	1.84 ± 0.07	0.001

^1^ BBW—bled body weight; ^2^ Other organs—trachea, lung, heart, liver, diaphragm, kidneys, spleen.

**Table 2 animals-13-01082-t002:** Seasonal variation in the proportion of edible internal organs in the bled body weight (LSM ± SE).

	Season		
Item	Summer	Winter	*p*-Value
Liver, kg	0.823 ± 0.109	0.842 ± 0.112	0.896
Liver, % BBW	1.97 ± 0.24	2.06 ± 0.25	0.697
Kidneys, kg	0.112 ± 0.026	0.119± 0.029	0.912
Kidenys, % BBW	0.27 ± 0.04	0.29 ± 0.06	0.854
Heart, kg	0.388 ± 0.042	0.379 ± 0.044	0.867
Heart, % BBW	0.93 ± 0.16	0.92 ± 0.16	0.988
Tongue, kg	0.147 ± 0.015	0.158 ± 0.016	0.798
Tongue, % BBW	0.35 ± 0.07	0.39 ± 0.08	0.826

BBW—bled body weight.

**Table 3 animals-13-01082-t003:** Seasonal variation in the proportion of carcass parts in the cold carcass weight (LSM ± SE).

	Season		
Item	Summer	Winter	*p*-Value
Leg, kg	8.87 ± 0.39	9.41 ± 0.41	0.071
Leg, % CCW ^1^	39.23 ± 0.52	40.11 ± 0.51	0.178
Loin, kg	4.98 ± 0.32	5.18 ± 0.29	0.194
Loin, % CCW ^1^	22.04 ± 0.30	22.09 ± 0.38	0.843
Shoulder, kg	4.30 ± 0.26	4.43 ± 0.27	0.896
Shoulder, % CCW ^1^	19.01 ± 0.28	18.87± 0.26	0.869
Ribs with flank, kg	3.06 ± 0.23	3.07 ± 0.19	0.992
Ribs with flank, % CCW ^1^	13.53 ± 0.19	13.02 ± 0.21	0.369
Neck, kg	1.40 ± 0.15	1.39 ± 0.15	0.783
Neck, % CCW ^1^	6.19 ± 0.28	5.91 ± 0.26	0.492

^1^ CCW—cold carcass weight.

**Table 4 animals-13-01082-t004:** Seasonal variation in the composition of carcass and carcass cuts.(LSM ± SE).

Item	Summer	Winter	*p*-Value
Carcass			
Lean, kg	17.36 ± 0.23	17.38 ± 0.43	0.984
Lean, %	76.74 ± 0.31 ^a^	74.07 ± 0.32 ^b^	0.032
Fat, kg	0.86 ± 0.11 ^B^	1.54 ± 0.12 ^A^	0.001
Fat, %	3.79 ± 0.25 ^B^	6.55 ± 0.24 ^A^	0.001
Bones, kg	4.41 ± 0.18	4.55 ± 0.18	0.965
Bones, %	19.47 ± 0.62	19.38 ± 0.61	0.896
Leg			
Lean, kg	6.88 ± 0.36	7.08 ± 0.36	0.692
Lean, %	77.57 ± 0.79 ^a^	75.23 ± 0.78 ^b^	0.039
Fat, kg	0.33 ± 0.11	0.59 ± 0.11	0.124
Fat, %	3.75 ± 0.28 ^B^	6.23 ± 0.27 ^A^	0.002
Bones, kg	1.66 ± 0.15	1.74 ± 0.15	0.766
Bones, %	18.68 ± 0.65	18.54 ± 0.64	0.794
Loin			
Lean, kg	3.63 ± 0.28	3.63 ± 0.28	0.996
Lean, %	72.88 ± 0.72 ^a^	69.96 ± 0.73 ^b^	0.016
Fat, kg	0.21 ± 0.06	0.39 ± 0.06	0.063
Fat, %	4.29 ± 0.38 ^B^	7.63 ± 0.39 ^A^	0.003
Bones, kg	1.14 ± 0.11	1.16 ± 0.11	0.901
Bones, %	22.83 ± 0.32	22.41 ± 0.33	0.637
Shoulder			
Lean, kg	3.36 ± 0.22	3.29 ± 0.22	0.862
Lean, %	78.27 ± 0.74 ^b^	75.32 ± 0.74 ^a^	0.038
Fat, kg	0.12 ± 0.03 ^B^	0.24 ± 0.03 ^A^	0.001
Fat, %	2.81 ± 0.29 ^B^	5.53 ± 0.29 ^A^	0.001
Bones, kg	0.81 ± 0.09	0.83 ± 0.09	0.859
Bones, %	18.92 ± 0.35	19.15 ± 0.35	0.629
Ribs with flank			
Lean, kg	2.43 ± 0.19	2.15 ± 0.19	0.659
Lean, %	79.35 ± 0.82 ^a^	76.06 ± 0.82 ^b^	0.021
Fat, kg	0.16 ± 0.04 ^b^	0.24 ± 0.04 ^a^	0.034
Fat, %	5.42 ± 0.41 ^B^	8.63 ± 0.41 ^A^	0.001
Bones, kg	0.47 ± 0.06	0.43 ± 0.07	0.598
Bones, %	15.23 ± 0.24	15.31 ± 0.24	0.629
Neck			
Lean, kg	1.05 ± 0.11	0.93 ± 0.11	0.842
Lean, %	74.79 ± 0.65	73.12 ± 0.65	0.758
Fat, kg	0.02 ± 0.001 ^B^	0.04 ± 0.001 ^A^	0.001
Fat, %	1.68 ± 0.22 ^B^	3.46 ± 0.22 ^A^	0.001
Bones, kg	0.33 ± 0.05	0.30 ± 0.05	0.796
Bones, %	23.53 ± 0.29	23.42 ± 0.29	0.862

^A, B (a, b)^ Values in rows with different superscripts differ significantly at *p* < 0.01 (*p* < 0.05).

**Table 5 animals-13-01082-t005:** Seasonal variation in the physicochemical traits of edible organs from the wild fallow deer (LSM ± SE).

Season		Organs				Effect (*p*-Value)			
		Liver	Kidneys	Heart	Tongue	S	O	T	Interaction
pH_24_	Summer	5.82 ± 0.04 ^A^	6.23 ± 0.04 ^B^	6.18 ± 0.04 ^B^	5.66 ± 0.04 ^A^	0.066	<0.0001	0.001	S*O*T 0.071
pH_48_		5.84 ± 0.04 ^A^	6.25 ± 0.04 ^B^	6.19 ± 0.04 ^B^	5.65 ± 0.04 ^A^				
pH_24_	Winter	5.77 ± 0.04 ^A^	6.24 ± 0.04 ^B^	6.17 ± 0.04 ^B^	5.63 ± 0.04 ^A^				
pH_48_		5.78 ± 0.04 ^A^	6.25 ± 0.04 ^B^	6.16 ± 0.04 ^B^	5.62 ± 0.04 ^A^				
Moisture (%)	Summer	79.23 ± 0.31 ^A^	80.95 ± 0.31 ^B^	72.52 ± 0.31 ^C^	68.13 ± 0.31 ^D^	0.158	<0.0001	-	S*O 0.364
	Winter	78.94 ± 0.33 ^A^	80.12 ± 0.33 ^B^	72.06 ± 0.33 ^C^	68.42 ± 0.33 ^D^				
WHC (%)	Summer	31.04 ± 0.73 ^A^	27.47 ± 0.73 ^B^	34.08 ± 0.73 ^C^	19.71 ± 0.73 ^D^	0.183	<0.0001	-	S*O 0.182
	Winter	30.62 ± 0.78 ^A^	27.17 ± 0.78 ^B^	33.81 ± 0.78 ^C^	19.97 ± 0.78 ^D^				
Plasticity (cm^2^)	Summer	4.23 ± 0.18 ^A^	2.89 ± 0.18 ^Ba^	3.76 ± 0.18 ^C^	2.16 ± 0.18 ^Bb^	0.643	<0.0001	-	S*O 0.764
	Winter	4.16 ± 0.19 ^A^	2.77 ± 0.19 ^Ba^	3.59 ± 0.19 ^C^	2.14 ± 0.19 ^Bb^				

^A, B, C, D (a, b)^ Values in rows with different superscripts differ significantly at *p* < 0.01 (*p* < 0.05); effects of fixed factors: S—season; O—internal organ; T—time; interactions between factors: S*O*T, S*O.

**Table 6 animals-13-01082-t006:** Seasonal variation in the chemical composition of edible internal organs from the wild fallow deer (LSM ± SE).

Item	Season	Organs				Effect (*p*-Value)	
		Liver	Kidneys	Heart	Tongue	Season	Organ
Dry matter (%)	Summer	20.77 ± 0.31 ^A^	19.05 ± 0.31 ^B^	27.48 ± 0.31 ^C^	31.87 ± 0.31 ^D^	0.158	<0.0001
	Winter	21.06 ± 0.33 ^A^	19.87 ± 0.33 ^B^	27.94 ± 0.33 ^C^	31.58 ± 0.33 ^D^		
Crude protein ^1^	Summer	17.84 ± 0.37 ^A^	15.97 ± 0.37 ^B^	20.80 ± 0.37 ^C^	16.44 ± 0.37 ^B^	0.425	<0.0001
	Winter	17.79 ± 0.39 ^A^	16.07 ± 0.39 ^B^	20.69 ± 0.39 ^C^	16.17 ± 0.39 ^B^		
Extractable fat ^1^	Summer	1.38 ± 0.12 ^A^	1.26 ± 0.12 ^A^	1.53 ± 0.12 ^A^	13.86 ± 0.12 ^B^	0.002	<0.0001
	Winter	1.71 ± 0.11 ^A^	1.76 ± 0.11 ^A^	1.57 ± 0.11 ^A^	14.26 ± 0.11 ^B^		
Ash ^1^	Summer	1.31 ± 0.03 ^A^	1.26 ± 0.03 ^A^	1.11 ± 0.03 ^B^	1.01 ± 0.03 ^C^	0.168	0.019
	Winter	1.33 ± 0.02 ^A^	1.29 ± 0.02 ^A^	1.13 ± 0.02 ^B^	1.04 ± 0.02 ^C^		
W/P ^2^	Summer	4.45 ± 0.13 ^A^	5.09 ± 0.13 ^B^	3.49 ± 0.13 ^C^	4.15 ± 0.13 ^A^	0.601	<0.0001
	Winter	4.24 ± 0.13 ^A^	5.10 ± 0.13 ^B^	3.59 ± 0.13 ^C^	4.24 ± 0.13 ^A^		

^A, B, C, D^ Values in rows with different superscripts differ significantly at *p* < 0.01; ^1^ Expressed as percentage of dry matter; ^2^ W/P—water to protein ratio.

## Data Availability

Data available on request.

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
