# Peer review of "Seasonal Variation in the Body Composition, Carcass Composition, and Offal Quality in the Wild Fallow Deer (Dama dama L.)"

_animals, 2023, doi:10.3390/ani13061082_

Round 1

Reviewer 1 Report (Previous Reviewer 1)

The manuscript was revised. I recommend it for further publishing stage.

Author Response

Dear Reviewer.

I appreciate your time and input in the review process.

Best regards,

Reviewer 2 Report (Previous Reviewer 4)

The paper has been improved according to my suggestions. Thus, the manuscript must be accepted in the current form

Author Response

Dear Reviewer.

I appreciate your time and input in the review process.

Best regards,

Reviewer 3 Report (New Reviewer)

Dear authors, this paper deals with the seasonal variation in the body composition, carcass composition, and offal quality in the wild fallow deer. Interesting topic and out of the normally treated subject. However, I have some concerns, listed in the attached file

Author Response

Dear Reviewer.

Thank you for the suggestions and comments. I have corrected the articles according to Your guidelines. All the changes are marked in red, thought the review was in form of comments in a pdf document I have tried to address these comments below.

- I have not added the suggested citation 10.1080/1828051X.2022.2032850 because from my point of view discussing wild ungulates’ condition with domestic dairy cattle is not proper, and it does not fit in the scope of the revised article.

- I have changed the place of the ‘disclaimer’ by placing it in the Discussion section and moved the pointed part of the Introduction to the Materials and Methods

- The body weight of animals has been added – line 73.

- Citation – Serrano et a., 2020 was changed to Serrano et al. [18]

Best regards,

Round 2

Reviewer 3 Report (New Reviewer)

Dear authors, thank you for your revisions, however, I would like to highlight that the dynamics with animals change their body composition and proportions are the same in all the animals, independently if whether they are domestic or not.

Best regards

Author Response

Thank you very much

This manuscript is a resubmission of an earlier submission. The following is a list of the peer review reports and author responses from that submission.

Round 1

Reviewer 1 Report

The topic is interesting, the manuscript contains enough data to generate interesting and relevant discussion on the studied topic. However, certain improvements should be made.

In my opinion, the authors studied the body composition (not the slaughter value) of the fallow deer. Therefore, appropriate corrections should be done in the title of the manuscript, title of Table 1 and in the text. There are some other strange terms in the text.  The terms moisture and water holding capacity (WHC) are widely used to describe meat composition and properties and could, therefore, be also used instead of total water and free water contents (Lines 82-83, 129,131-132, Table 5) to describe the composition and properties of the offal. It is also unclear why the authors use sophisticated “point of measurement” instead of different tissues of the offal.

Some results from the study should be added to Simple Summary without repeating Abstract text.

Instead of listing all variables in the description of the statistical analysis (Lines 99-107), it would be better to specify the applied models, including fixed factors, interactions for body and carcass composition, offal physiochemical characteristics and chemical composition analyses. The text (Lines 107-109) about the fact that there were no significant interactions would be more appropriate in the Results section after the description of the effects of the factors studied than in the Methods.

The headings of the columns in the tables are unclear, so I would like to suggest you to consider whether it would be better to place some tables in portrait-oriented pages with two rows for each variable (for weight and percentage in body and carcass).

If the authors decide to leave the current tables, it should be indicated that the weight is expressed in kg (alone “kg” does not indicate anything). Also, two columns on the right side of Table 1 are without the headings. I guess there are p-Values, but p-Values do not show kg or %, they show the effects of season and other factors. This should be corrected. Since all the traits are presented in the expression of weight and proportions, it would be more appropriate to highlight the differences and similarities of the seasonality impact on the characteristics according to their expression in the text.

The season affected only the fat content of the offal (Table 6), but not the overall chemical composition as written in the Conclusions (Lines202-203).

Reviewer 2 Report

Dear Authors,

the paper is correct prepared, however I have a few mistakes to change. I will describe them line by line:

3 - add L. 175 into (Dama dama)

4 - Ludwiczak instead of Ludwiczaka, right?

5 - Department instead of aDepartment

37 - empty line, please correct

44 and 60 - put the same style of bracket ( ...) instead of [...)

73, 77 - °C instead of oC

84, 89 - I would recommend deleting a repetition here. Modify it somehow, not to repeat this information two times so close

114 - too big space between: Table 1.     A

129 - was instead of is

136 - Table 1 - 6th and 7 th column - there is a lack of proper columns names? "Winter" should be above 4 and 5 column? Something is wrong in a first record.

Table 6 - 1 record - Tongue (together), not Tongu   and e in another record

151, 162 and others - numbers in brackets without space, e.g. [10,12]

165 - in wild Sika deer change into 'in wild sika deer'. The same in 184 line.

251 - something is wrong here: Foods. 2020 Jul 14;9(7):923.

278 - put a '197' inside the previous line (option in Microsoft Word).

Reviewer 3 Report

The manuscript is publishable but needs important improvements for the clarification of the results, especially the interpretation.

The introduction needs to put the information in the right context, namely farming and hunting practices in the country. For the research to be relevant, it is important to understand the system. Is farming culling all year round? What´s the common culling age? Is hunting allowed all year round? Is it common in July? Etc.

L59-60. How was the age determined? Was it 30 months both for the animals hunted in July and December?

L73, L77, L88. Please, use the typography correctly.

L79. The pH of what was measured?? Missing information.

L83. Where is the section 2.2.3?

The mixed models use for independent variables (season, PM, postmortem time and interactions – not clear how many interactions were tested). This should be better explained and probably re-analysed. These are too many variables for a n=19.

The discussion needs important improvements. Main one, it is not possible to have a one-paragraph discussion. Too many ideas simultaneously, impossible to follow adequately.

L149. The authors should be very cautious about the results involving body weight. The animals used in the study are still growing. Didn´t reach the adult weight yet. And that explain all the differences (or lack of differences, see a further comment on it) involving body weight. Thus, not much relevant for discussion.

L153-156. The patter described in the cited reference is just the opposite as the one found by the authors! Please, reconsider the idea and rewrite the sentence.

L156-161. The explanation is correct. But please, be even more clear stating that the main difference is due to a very full rumen in summer and a not so full one in winter.

L167-168. Which are the implications of this for the manuscript?

L174-176. I would suggest to the authors to reconsider this sentence for cervids. They usually do not accumulate fat in summer, but use the resources for supporting reproduction and antler growth, and getting ready (especially males) for the coming rut season. After that is when fat storing actually starts, getting enough resources for the not-growing winter period (well explained in the manuscript).

L184-184. This also needs further thinking. The reference is about subcutaneous fat, but the current manuscript is about internal fat. Please, notice that the fat deposition pattern can be different for different body tissues. See for example Ceacero et al., 2020 (DOI 10.1017/S1751731119002325).

Continuing with the previous comment, I missed any discussion about the intramuscular fat, which seems to follow a similar pattern as the internal fat.

L192-193. Nice speculation, but the true is much more simple: young animals are still growing, which compensates the weight loss common in adults. So easy.

Conclusions. The authors should be more accurate with the results. Indeed, the significant results are very few. From L210-211, it may be concluded wide results, but these were significant just for fat.

I would suggest changing the title according to the results. Indeed, there is no information in the paper about slaughter value, the results about carcass composition are mainly due to the age, not relevant, and there are no significant results about offals (moreover, just about offal production, but not quality). On the other hand, there is no information about fat, which is the only part offering significant results. Altogether, the title is nice but do not reflect at all the results obtained.

Reviewer 4 Report

The manuscript is interesting scientific contributions to the knowledge of the seasonal variation in the slaughter value, carcass composition and offal quality in the wild fallow deer (Dama dama). According to the available research, the seasonal changes in the body condition of wild animals are related to the photoperiod and feed availability, with the second factor strictly dependent upon the type of environment and population density. In this regard, the aim this study was to evaluate the seasonal differences in the slaughter value, carcass composition, and offal quality in the wild fallow deer. The paper has high scientific level, the experiment is well designed, the discussion is consistent and the final conclusions are interesting. Therefore, the manuscript may be published in Animals making major revision:

Suggestions for edition as well as some comments are the following:

Abstract

Please rewrite the abstract giving more information about the obtained results

Keywords

Please change these keywords “fallow deer, season, slaughter value, offal” because they are presented in the title.

Introduction

Please introduce some information about seasonal variation on the meat quality of deer meat. I suggest author to read this paper:

Serrano, M. P., De Palo, P., Maggiolino, A., Moure, M. P., Martínez, L. G., Domínguez, R., ... & Rodríguez, J. M. L. (2020). Seasonal variations of carcass characteristics, meat quality and nutrition value in Iberian wild red deer. Spanish journal of agricultural research, 18(3), 16.

In addition, please introduce more information about the carcass and meat quality of deer meat. I suggest authors the following paper to complete the introduction section:

Lorenzo, J. M., Maggiolino, A., Gallego, L., Pateiro, M., Serrano, M. P., Domínguez, R., ... & De Palo, P. (2019). Effect of age on nutritional properties of Iberian wild red deer meat. Journal of the Science of Food and Agriculture, 99(4), 1561-1567.

Kudrnáčová, E., Bartoň, L., Bureš, D., & Hoffman, L. C. (2018). Carcass and meat characteristics from farm-raised and wild fallow deer (Dama dama) and red deer (Cervus elaphus): A review. Meat science, 141, 9-27.

Kudrnáčová, E., Bartoň, L., Bureš, D., & Hoffman, L. C. (2018). Carcass and meat characteristics from farm-raised and wild fallow deer (Dama dama) and red deer (Cervus elaphus): A review. Meat science, 141, 9-27.

Serrano, M. P., Maggiolino, A., Landete-Castillejos, T., Pateiro, M., Barbería, J. P., Fierro, Y., ... & Lorenzo, J. M. (2020). Quality of main types of hunted red deer meat obtained in Spain compared to farmed venison from New Zealand. Scientific Reports, 10(1), 1-9.

Please update references. There are a lot of recent studies about deer meat. Please include references before 2017

Material and methods

Line 92, 128, post-mortem (in italic)

Results

Table 1 is not correct. Please revise the first line of table (regarding summer, winter and Effect (p-value) according to Table 2 (correct).

Table 6, please express the results as dry matter.

Discussion

The results are well discussed, however, the authors used old references. Try to re-write the discussion with recent studies.

I hope that my comments can improve the manuscript.